# Innovation Strategy for Green Development and Carbon Neutralization in Guizhou—An Overview

Jun Yan [1,2], Wu Yang [1,2,*] , Zhang Min [1,2] and Mingxing Yang [1,2]

1 Faculty of Resources and Environmental Engineering, Guizhou Institute of Technology, Guiyang 550003, China
2 Engineering Research Center of Carbon Neutrality in Karst Areas, Ministry of Education, Guiyang 550003, China
* Correspondence: yangwu@git.edu.cn

**Abstract:** A carbon peak in 2030 and carbon neutrality in 2060 are major strategic development goals for China. Driven by the ambitious goal of achieving a carbon peak and carbon neutrality, the development of green innovation technology is an important method of achieving these aims. Speeding up energy transformation and tapping into the carbon sink capacity of the natural ecosystem are key to this process. The strategic path of green development deserves further discussion. This study takes Guizhou province as an example. Based on the actual situation of Guizhou province and the characteristics of karst areas, through the collection and collation of the existing literature, policies, and technologies and the analysis of typical cases, this paper summarizes and analyzes ecological restoration and negative carbon emissions in karst areas; water-energy-carbon coupling, energy saving, and emissions reduction technologies; industrial energy saving and emission-reduction technologies in karst areas; and CCUS technology for carbon dioxide capture, utilization, and storage. On this basis, the trend and orientation of green development in Guizhou are studied and judged, and countermeasures such as adhering to clean and efficient low-carbon utilization, strengthening the research on and development of carbon emission-reduction technology, and implementing carbon sink capacity buildings are put forward. Key core technology research and development innovation are recommended to establish a low-carbon science and technology innovation system. The efficient use of energy, the recycling of resources, negative emissions, and other strategies should be promoted. We also posit specific suggestions such as accelerating the transformation and application of green and low-carbon scientific and technological achievements.

**Keywords:** innovation strategy; carbon sink; sustainability; Guizhou; suggestion

## 1. Introduction

In order to tackle the climate change crisis and achieve the temperature control goals of the Paris Agreement, China has made solemn commitments to achieve peak carbon emissions by 2030 and carbon neutrality by 2060, raise the proportion of non-fossil energy from 20% to 25%, and increase its forest stock from 4.5 billion cubic meters to 6 billion cubic meters. Achieving these carbon peak and carbon neutrality goals will profoundly affect the development and utilization of China's existing energy resources, and we also need to explore and discover more ways to absorb carbon dioxide.

Guizhou is located in the middle of China's karst area, and its natural environment and social development are typical of this area. The karst distribution area accounts for 73.79% of the total land area of the province. Guizhou is also an important coal and phosphate industry base in China, which are key contributors to carbon emissions. Driven by the ambitious goal of a carbon peak and carbon neutrality based on the characteristics of karst areas, this study summarizes domestic and foreign ecological restoration and negative carbon emissions; water-energy-carbon coupling, energy saving, and emissions reduction; industrial energy saving and emissions reduction technologies in karst areas; and carbon

dioxide capture, utilization, and storage (CCUS) technologies. Starting from the idea of green development, on the basis of a systematic analysis of the advantages, disadvantages, opportunities, and challenges of Guizhou's medium- and long-term development, this paper studies and judges the trend and orientation of Guizhou's green development and combines this with the goal of carbon neutrality. This paper puts forward some counter-measures, such as insisting on clean and efficient low-carbon utilization, strengthening the research on and development of carbon emission-reduction technologies, and increasing the carbon sink capacity, and puts forward some specific suggestions, such as strengthening top-level design, defining the path of emission reduction, guiding the transformation of enterprises, and shoring up scientific and technological talents. It is hoped that these strategic proposals will aid Guizhou's high-quality development and help it to achieve the double carbon goal as soon as possible (Figure 1) [1–5].

| | 11th FYP (2006-2010) | 12th FYP (2011-2015) | 13th FYP (2016-2020) | 14th FYP (2021-2025) |
|---|---|---|---|---|
| General innovation approach | Ramp up technology manufacturing to boost exports. | Prime domestic markets and manufacturing innovations. | Seek novel innovations in priority technology areas. | Keep edge in manufacturing and prime breakthrough innovations. |
| Key focus areas for energy innovation | Nuclear, coal, automobiles and new materials. | Solar, wind, electric vehicles and charging. | Next-generation renewables, energy storage, new energy vehicles and batteries, smart power grids and buildings energy efficiency. | Next-generation batteries and new energy vehicles, hydrogen and fuel cells, advanced biofuels, CCUS and smart digital systems. |

**Figure 1.** Technology development and key energy innovation priorities outlined in China's recent five-year plans.

## 2. Status and Prediction of Carbon Emissions in Guizhou Province

According to statistics released by the Guizhou Energy Bureau, the total energy consumption of the province in 2020 was 106 million tons of standard coal. Coal consumption was 119 million tons, electricity consumption was 158.6 billion kilowatt-hours, refined oil consumption was 6.83 million tons, and natural gas (including coal mine gas) consumption was 3.1 billion cubic meters. In 2020, the installed capacity of non-fossil energy accounted for 52.9% of the total, which was 8.1 percentage points higher than the national average; the consumption of non-fossil energy accounted for 17.4% of the total, which was 1.7 percentage points higher than the national average; coal consumption accounted for 69.1% of the total, which was 12.3 percentage points higher than the national average; energy consumption per unit of GDP was 0.6 tons of standard coal/10,000 CNY, which was 22 percentage points higher than the national average. The proportion of coal consumption and the energy consumption intensity in Guizhou are still high, and there is much room for improvement. The carbon dioxide emissions in Guizhou's energy sector in 2020 amounted to about 220 million tons (excluding 32 million tons of carbon emissions from power transmission). The carbon emissions from coal consumption amounted to about 196 million tons. Carbon dioxide emissions in the energy sector were mainly coal consumption emissions, which accounted for 89% of the total (12 percentage points higher than the national average), while power generation emissions from coal consumption accounted for 54.5% of the total, which was comparable to the national average. Carbon dioxide emissions per unit of GDP

were 1.2 tons/10,000 CNY, which was 1.2 times the national average. The coal consumption of the coal-fired power supply was 323 g per kilowatt-hour, which was 17 g higher than the national average.

### 2.1. Economic Development Trends and Total Energy Demand in Guizhou

According to the strategic plan for building a modern socialist country in an all-round way, China aims to realize socialist modernization by 2035. It is predicted that China's GDP will double between 2020 and 2035, and it is expected to double again between 2035 and 2060. According to statistics released by the Guizhou Provincial Bureau of Statistics, it is predicted that the total GDP of Guizhou in 2035 will be about 5 trillion CNY, and the per capita GDP will be roughly equal to the national average; the total GDP in 2060 will be about 12 trillion CNY, and the per capita GDP will be 1.1 times the national average. According to the correlation between energy consumption and economic growth, the future energy consumption intensity of Guizhou is predicted via a fitting analysis using a logarithmic function model. According to the medium- and long-term forecasts of the economic development and energy consumption intensity of Guizhou province, it is estimated that the total energy demand of Guizhou province in 2025, 2030, 2035, 2050, and 2060 will reach 132, 156, 180, 237, and 275 million tons of standard coal, respectively. From 2020 to 2030, the average annual growth rate of GDP and total energy consumption in the province is expected to be 7.6% and 3%, respectively. From 2030 to 2060, the average annual growth rate of GDP and total energy consumption in the province is expected to be 3.9% and 1.4%, respectively. The total amount of energy has a strong supporting capacity for the sustained and rapid development of Guizhou's economy, and a safe and stable supply of energy is guaranteed, which is also an important prerequisite for the development of "carbon peak, carbon neutralization" in the field of energy.

### 2.2. Prediction of Carbon Emissions in Energy Sector of Guizhou Province

This study uses a scenario analysis to identify the factors that impact carbon emissions under different development scenarios in order to predict the level of carbon emissions in Guizhou province under various scenarios. We used three scenarios, namely a baseline scenario, a high-speed scenario, and a low-carbon scenario. They corresponded with the indicators of medium and high growth with positive regression coefficients. Then, according to the strategic policy interpretation of economic development and energy development in Guizhou province, the current situation of economic and social development and the development trend of energy structure in Guizhou province were clarified. The parameters of total population, urbanization rate, resident consumption level, total energy consumption, and per capita GDP of Guizhou province in the future under different development scenarios were predicted. Finally, the evolution trends of carbon emissions in Guizhou province in the future are predicted.

The high-speed scenario is characterized by "rapid growth and slow decline of carbon emissions". From 2020 to 2035, the total carbon emissions increase rapidly and reach a peak. The peak carbon emissions in 2035 are about 304 million tons. From 2035 to 2060, the total carbon emissions decrease slowly, the carbon neutralization cycle is long, and the total carbon emissions in 2060 are about 221 million tons.

The baseline scenario reflects the characteristics of "carbon peak acceleration, rapid decline, and neutralization". From 2020 to 2030, the total carbon emissions increase rapidly and reach a peak, with the peak value being about 282 million tons in 2030. From 2030 to 2050, the total carbon emissions decrease rapidly, reaching around 55% of the peak value by 2050. Further carbon emissions reduction is achieved in 2050–2060, and the total amount of carbon emissions in 2060 are about 131 million tons. With the application of large-scale carbon capture and utilization technology, the energy sector in Guizhou can achieve the goal of carbon neutralization by 2060.

The low-carbon scenario reflects the characteristics of "carbon peak acceleration, rapid decline to achieve neutralization". In 2030, the peak value of carbon emissions in Guizhou

are about 282 million tons. From 2030 to 2050, the total carbon emissions decrease rapidly, and the total carbon emissions in 2050 are about 30% of the peak value. From 2050 to 2060, carbon emissions are further reduced, and the total carbon emissions in 2060 are about 56 million tons. The ecological carbon sink capacity of Guizhou is fully utilized, and the carbon neutralization target in the energy sector is reached by 2060.

### 3. Overview of Carbon Neutralization Innovation Technology

Guizhou has the largest wet phosphoric acid purification chemical enterprise in China, which has considerable economic benefits. However, the energy consumption rate of the phosphorus chemical industry is increasing at a rate of 10% annually. As a carbon-based energy source, the problem of $CO_2$ emissions from phosphorus chemical products should not be underestimated, especially in the process of raw material pretreatment and the process of waste and energy consumption. In the non-ferrous metal industry, especially the aluminum industry, there is also high energy consumption and pollution in the production process, which has become a bottleneck in the context of achieving the goal of "double carbon". According to official statistics, coal power generation in Guizhou is still increasing. It reached 115.87 billion kWh in 2019, an increase of 11.1% over the same period in the previous year. It accounts for more than 60% of the power generation in Guizhou's power grid. With a decline in phosphorus coal and bauxite, the increase in energy and material consumption increases the pressure to reduce carbon emissions. The development of resource recycling technology and a reduction in energy consumption and waste emissions in industrial processes also provides paths for "carbon neutralization". Therefore, based on new technologies, new materials, new equipment, and new industrial production processes, we can minimize the pollution caused by production and fundamentally solve the problem of green development in order to maintain the ecological bottom line and promote carbon reduction and carbon neutralization.

At the same time, Guizhou is the province with the largest area of karst rocky desertification, which is the key reason for its ecological degradation, poverty, and backward development. The weathering of carbonate rocks is the main geological factor that causes rocky desertification. Over the past two decades, the state has invested a lot of resources in the ecological restoration of areas affected by karst rocky desertification and has achieved great results. At present, Guizhou is one of the provinces with the highest rates of vegetation coverage in southwest China. Through long-term afforestation, the southwest forest area, which is dominated by Guizhou, can create 350 million tons of carbon sinks every year, accounting for 31.5% of the national terrestrial ecological carbon sinks. In this way, it can make outstanding contributions to ecological carbon sequestration.

In recent years, karst carbon sinks have become an important type of carbon sink. They play a buffer role in the change of global atmospheric $CO_2$ concentration. However, due to a lack of relevant theoretical research, they have been neglected for a long time. The distribution area of carbonate rocks in China is about 3.446 million square kilometers, accounting for one third of the total land area. The karst carbon sink flux is considered to be of the same order of magnitude as the forest carbon sink. As early as the fifth report of the Intergovernmental Panel on Climate Change in 2013, it was made clear that the dissolution process of carbonate rocks, as a technical method of removing $CO_2$, has become an important means to cope with climate change, alongside terrestrial ecological processes, marine carbon sinks, and artificial direct capture. The southwest karst area, with Guizhou at its center, is the largest carbonate rock distribution area in the world and has huge carbon sink potential. Investigating the karst carbon cycle in karst areas; identifying the formation conditions, influencing factors, and migration and transformation patterns of the karst carbon cycle; calculating the source and sink; and evaluating the karst carbon sink flux will all contribute to strategic decision-making in the context of "carbon peak, carbon neutral" in China [6–10].

### 3.1. Water-Energy-Carbon Coupling

With the rapid development of urbanization and industrialization, water and energy have gradually become the main limiting factors for development. A large amount of water is used for energy supply (coal mining, power generation, etc.), and energy is needed in the processes of water supply, water use, and wastewater discharge. The tracking of inter- and intra-regional energy and water flows and the quantification of their interdependencies are fundamental to achieving energy and water balance. The concept of the water-energy-carbon relationship can be used to study the interdependence, coupling mechanisms, and interrelation between water and energy, and its conversion process is embedded in the multi-disciplinary chain of multi-scale interweaving. There have been many studies on the relationship between water, energy, and carbon, but the quantitative analyses in these studies mainly focus on their parallel relationships. Many studies have been conducted on water used for energy supply at home and abroad. In recent years, many scholars have used materials, material flow analysis, and life cycle assessment to evaluate the whole circulation process of substances or commodities and their mutual demand in order to determine the leverage points of related systems. To simulate the complex interaction and dynamics of the water-energy relationship in urban systems, a number of studies at home and abroad have concluded that water-related energy accounts for 13% of the total electricity and 18% of the natural gas used by people in their daily lives on average, and a system-oriented urban water-energy relationship network has been established. Its structural characteristics and sectoral dynamics have been analyzed, and urban relations have been integrated into a metabolic framework. Regarding important human advances in more practical and sustainable urban planning and management, input-output analysis can assess indirect flows in addition to direct flows in a complex system to calculate the quantities needed to produce goods and services based on intersectoral interactions and exchanges [11–17].

Guizhou is rich in precipitation resources and has an annual precipitation rate of about 1200 mm. However, it is located in the Yunnan–Guizhou Plateau, where the karst area is abundant and extensive and the engineering water shortage is very serious. The annual investment in the water conservancy industry is about 40 billion CNY, and there are many research units and enterprises in the Guizhou water conservancy industry. At present, there are about 200 enterprises in the water conservancy and hydropower engineering industry, including the Guizhou Water Conservancy and Hydropower Survey and Design Institute, the Guiyang Design Institute of China Hydropower Consulting Group, and the Guizhou Water Conservancy Research Institute, as well as other water conservancy and hydropower design institutes in various other prefectures and cities. These scientific research institutions and enterprises are mainly engaged in research on and the construction of hydrology, water resource, and water conservancy projects. No studies have been conducted on carbon neutrality in karst areas or in Guizhou province.

### 3.2. Industrial Energy Conservation and Emission Reduction in Karst Areas

Energy saving and emission reduction refers to reducing energy waste and reducing emissions. Energy is related to the national economy and people's livelihood and is an important foundation for economic and social development. Due to the rapid growth of the national economy and the acceleration of urbanization, the demand for energy is increasing. Therefore, we must speed up the adjustment and optimization of energy structures, vigorously cultivate new energy industries, incorporate carbon dioxide emission reduction targets as binding indicators into the medium- and long-term planning of national economic and social development, and focus on building industrial, transportation, and construction systems with low carbon emissions as the main priority.

Guizhou is rich in phosphorus, coal, and water energy, which is its natural endowment and its comparative advantage in terms of its economic and social development. Guizhou is thus a major province that transforms energy advantages into economic advantages, and, as such, it will become an important energy base in southern China. The limitation

and scarcity of energy resources, as well as the strategies and security surrounding them, determine that the comprehensive utilization of energy, energy conservation, and emission reduction must be supported by scientific and technological progress (Figure 2).

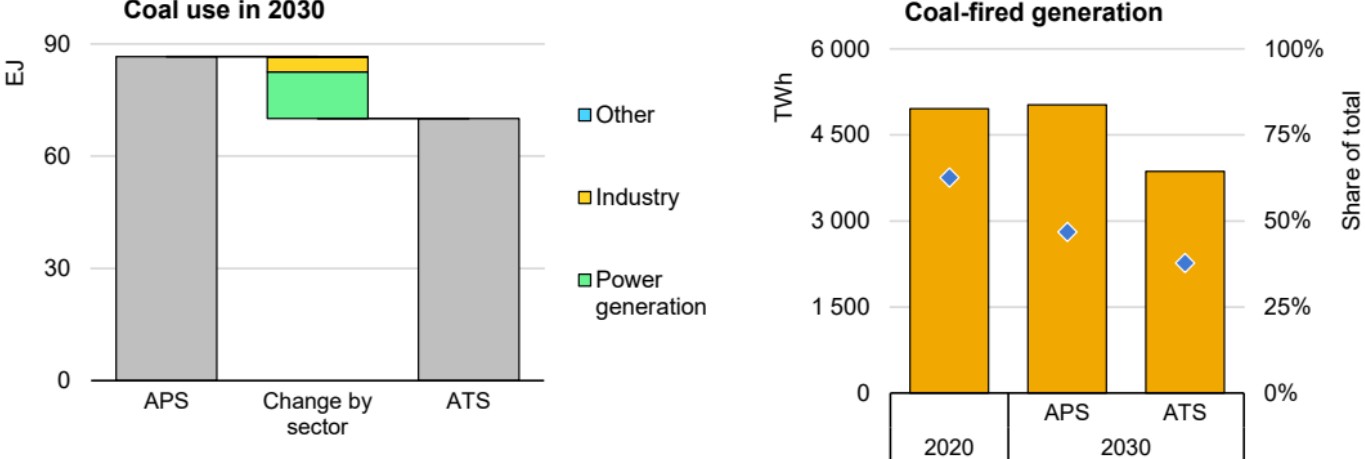

**Figure 2.** Total coal consumption in 2030 by scenario/case and coal-fired power generation in China under the APS and ATS (IEA 2021) [18]. Note: APS = announced pledges scenario, ATS = accelerated transition scenario.

3.2.1. Energy Saving and Emission Reduction in Phosphorus and Coal Chemical Enterprises

The phosphorus and coal chemical industries belong to the process industry that has more prominent carbon consumption and emissions rates, and the discharge of solid waste and waste liquid in the production processes are also more prominent, which has a greater impact on the weak karst environment.

There are predominantly two methods that are used in the phosphorus chemical industry. One is the wet method, which uses strong acid to decompose phosphate rock into wet-process phosphoric acid or multi-component phosphate fertilizer. Elemental phosphorus products, such as wet-process phosphoric acid, are then further processed into phosphate compound fertilizers and their by-products. Purified phosphoric acid is used to produce feed-grade calcium hydrogen phosphate, and industrial-grade phosphate, phosphogypsum, and fluorosilicon resources are used for recycling and deep processing. The other method is the thermal method, which involves reducing the pentavalent phosphorus in the raw material of phosphate rock into elemental phosphorus via carbon reduction at a high temperature and then using this to create industrial sodium tripolyphosphate (pentasodium) and its by-products, mainly for the production of yellow phosphorus, thermal phosphoric acid, pentasodium and its by-products, phosphorus furnace tail gas, and phosphorus slag for recycling and deep processing. Wet-process phosphoric acid (WPA) is almost entirely produced by a dihydrate process in the domestic phosphate fertilizer industry. The dihydrate process is a mature process that requires low levels of investment and involves a simple operation, but there are two fundamental problems that are difficult to solve. On the one hand, the concentration of phosphoric acid produced is limited to less than 30% (calculated by P2O5). On the other hand, the by-product dihydrate phosphogypsum has low utilization value and is piled up as industrial waste residue in many places. Processing it into building materials or re-roasting it into sulfuric acid consumes a lot of energy and causes secondary pollution. Kiln process phosphoric acid (KPA) is manufactured by the Western Petroleum Research Corporation (ORC). A new technology developed in the 1980s for the production of industrial phosphoric acid via the direct utilization of medium- and low-grade phosphate rock has the advantage that the heat released from the phosphorus oxidation process is used to provide the heat required for the reduction of phosphate rock so that the total energy consumption is greatly reduced.

In addition, research has been conducted on phosphogypsum treatment processes and technologies; processes and technologies for the recovery and comprehensive utilization of resources associated with phosphate rock; iodine extraction from fluorosilicic acid and iohexol; the preparation, storage, and utilization of hydrogen energy; the preparation and utilization of hydrogen-rich energy; the development and application of new solar energy conversion materials; and hydrogen-rich, raw-material fuel cells. These ideas are all based on diversifying of the energy supply and the development of new energy technologies. The key development direction involves reducing carbon emissions from the source.

Energy saving and emission-reduction technologies in the field of the new coal chemical industry mainly include ammonia synthesis technology, formaldehyde synthesis technology, coal chemical industry–coal power cogeneration technology, and methanol production technology. Ammonia, methanol, and other chemical raw materials are processed using coal resources, and the advanced treatment of coal tar is carried out in the synthetic ammonia industry, which can help effectively save energy and reduce emissions. In China's chemical industry, the consumption and use of formaldehyde derivatives are increasing day by day. The traditional production of formaldehyde derivatives is based on natural gas as a raw material, which is limited by the relatively small amount of natural gas resources in China, resulting in significantly higher production costs and a corresponding waste of resources in its production. In view of this situation, through the application of energy saving and emission-reduction technology for formaldehyde production, coal is used as a raw material to aid in the production and application of formaldehyde; that is, after the desulfurization treatment of coal, high-temperature compression is carried out by a compressor, and formaldehyde is generated via fine desulfurization system treatment to meet the production demand of formaldehyde. Through the effective use of coal resources, we can improve the efficiency of resource utilization to achieve better energy savings, reduced emissions, and increased production efficiency. Coal chemical co-production technology refers to the joint application of a variety of key technologies in the coal chemical industry to promote the comprehensive development and upgrading of its production processes. As a kind of energy-saving and emission-reduction technology, coal chemical co-production technology can effectively promote the utilization of material resources so as to obtain higher energy saving and emission-reduction benefits. For example, the combined application of liquefaction technology and coking technology provides more favorable technical conditions for coal chemical production, which is also an important area of research in the current development of the industry.

### 3.2.2. Energy Saving and Emission Reduction in Nonferrous Metals Industry

In recent years, technical research on making the non-ferrous metals industry low carbon and green has provided a scientific basis and engineering technical support for the transformation and upgrading of non-ferrous metals and other process industries towards sustainable economic development. Good progress has been made in providing support for green processes and carbon emissions reduction. However, due to the changeable and unpredictable properties of raw materials, as well as the complex physical and chemical processes involved in processing, it is difficult to describe the production process with accurate mathematical models, and the digitization problem has a long way to go before it can be solved. In contrast to the discrete manufacturing industry, the manufacturing processes of non-ferrous metals and other process industries generally have multiple coupled processes, and the global optimization of their overall operation is a mixed, multi-objective, multi-scale, dynamic conflict optimization proposition. The non-ferrous metal industry takes high efficiency, intelligence, and low-carbon processes as the theme of its manufacturing, realizes digitalization, intelligence, networking, and automation at the engineering technology level, and realizes high efficiency, greening, and safety at the enterprise manufacturing operation level. In the non-ferrous metal industry, aluminum is a metal with high energy consumption, carbon emissions, and bulk industrial solid waste output. For the seed decomposition of alumina, the mainstream method involves

using the aluminum technology that has been used for more than a hundred years. It utilizes complete mixing technology, which has high power consumption, high investment cost, and low product quality. "Micro-disturbance plug flow alumina seed decomposition technology" breaks through this recognition, realizes low energy consumption stirring and production in the production process, saves electricity significantly, and contributes to carbon reduction. Based on this technology, some similar technologies have been derived. To date, it has been used in the production of more than 10 million tons, and the prospect of its expansion to the industrial scale is being considered. However, the application of intellectualization, digitalization, and informatization still needs to be improved, and there is still room for improvement in terms of the equipment. In addition, the application of new materials and technologies is insufficient, necessitating further research.

### 3.3. CCUS Technology in Karst Areas

Carbon dioxide capture, utilization, and storage (CCUS) refers to the technical means of capturing and separating $CO_2$ from emission sources, such as energy utilization, industrial processes, or air, and transporting it to suitable sites for utilization or storage through tankers, pipelines, or ships so as to ultimately achieve $CO_2$ emission reduction. It is an indispensable part of China's technology portfolio for achieving a carbon peak and carbon neutrality. China's CCUS technology is comparable to the international advanced level as a whole, but the level of individual key technologies and commercial integration in carbon capture, transportation, and storage lags behind. $CO_2$ capture technology refers to the process of separating and enriching $CO_2$ from different emission sources by means of absorption, adsorption, membrane separation, low-temperature fractionation, and oxygen-enriched combustion. These are the bases and prerequisites for the development of CCUS technology. Transportation refers to the process of transporting the captured $CO_2$ to an available storage site, which is usually a tanker, ship, pipeline, etc. Generally, tank trucks are considered for small-scale and short-distance transportation, and pipeline transportation is preferred for long-distance, large-scale transportation or CCUS industrial clusters. Biological and chemical utilization technology $CO_2$ refers to the process of using different physical and chemical characteristics of $CO_2$ to produce products with commercial value and achieve emissions reductions. The level of technological development at home and abroad is basically synchronized, and it is at the stage of industrial demonstration as a whole. In the past decade, various biological and chemical utilization technologies have been developed. Some chemical utilization technologies in particular have made remarkable progress; the highest level of development is the synthesis of chemical materials from $CO_2$, such as the synthesis of organic carbonates, degradable polymers, and cyanate ester/polyurethane and the preparation of polycarbonate/polyester materials. China has made great progress in CCUS integration and optimization technology in the past decade. CCUS-integrated optimization technology abroad is generally at the stage of commercial application, while the development of related technologies in China still lags behind. In particular, pipeline network optimization and cluster hub technologies are only in the pilot stage. The development level of the above key technologies is not enough to support research on CCUS integration coupling and optimization technology in China, which restricts the development of large-scale CCUS demonstration projects. The lack of large-scale, full-chain integration demonstration projects further limits the improvement of integration optimization technology (Figure 3).

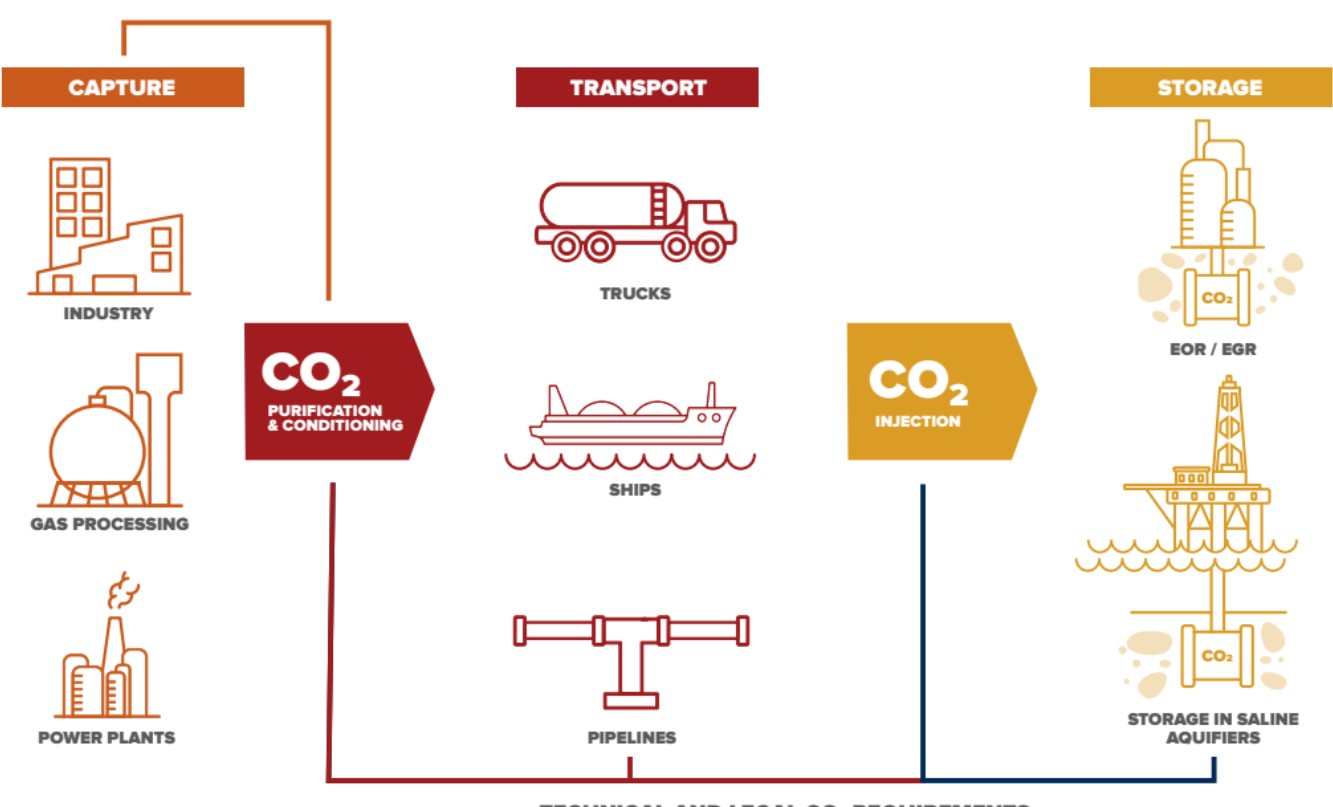

**Figure 3.** Carbon capture and storage—a conceptual diagram [11].

### 3.3.1. Study on New Dry Coal Powder Gasification Technology (CO$_2$)

Located in the Yunnan–Guizhou Plateau, Guizhou is in the middle of the Yunnan–Guizhou Base, a large coal base planned by the state that has a coal-bearing area of 70,000 km$^2$ and accounts for more than 40% of the total area of the province. The proven coal reserves rank fifth in the country and contain extremely rich coal resources, known as the "Jiangnan Coal Sea". Due to the energy structure of "rich in coal but insufficient oil and gas", coal is still the main energy source in China, and the generation, capture, and conversion of carbon dioxide cannot be avoided, Thus, in order to solve the problems of high pollution, high carbon emissions, and low energy efficiency in the process of coal conversion, we must realize the efficient, clean, and low-carbon utilization of coal resources in China, as well as the sustainable development of the energy supply in China. It is very important to achieve the goal of "carbon peak and carbon neutralization". As the key technology of clean coal power generation and the coal chemical industry, dry pulverized coal pressurized gasification technology has the characteristics of low specific oxygen consumption, high cold gas efficiency, low power consumption, and low cost of purification system and gas cooler. Shell gasification technology, Prenflo gasification technology, and GSP gasification technology are the main dry coal gasification technologies that have been industrialized in foreign countries. The technologies of pressurized entrained flow coal gasification with dry pulverized coal as a raw material in China mainly include multi-nozzle, opposed dry pulverized coal pressurized gasification technology, two-stage dry pulverized coal gasification technology, and space furnace gasification technology. In traditional dry pulverized coal pressurized gasification technology, N$_2$ is used as the carrier gas of pulverized coal in the furnace, resulting in a high proportion of N$_2$ in the syngas and a low calorific value of the gas, which is unfavorable to the subsequent CO$_2$ capture; at the same time, as an inert gas, N$_2$ does not participate in the reaction but absorbs heat, thus reducing the temperature in the furnace. Replacing N$_2$ with CO$_2$ as the carrier gas of the pulverized coal is a feasible solution, but this would increase the amount of CO$_2$

in the gasifier, affect the diffusion of $O_2$ and the coke reaction characteristics, and affect the gasification characteristics of the gasifier. At the same time, with an increase in the $CO_2$ level, the coke-$CO_2$ reaction becomes more important. In the southwest region, dry pulverized coal gasification technology uses the "three high coals" (high ash, high ash melting point, high sulfur). As a raw material, due to the poor activity, high ash melting point, high ash content, low volatile matter, and poor activity of the "three high coals", the conversion rate of the coal is low, the carbon content of the ash is high, and the effective components in the gas are low. Therefore, it is of great significance to develop a new dry pulverized coal gasification technology with the "three high coals" in southwest China as a raw material and $CO_2$ as a gasifying agent in order to achieve "carbon peak and carbon neutralization".

### 3.3.2. Research on $CO_2$-CCUS Materials of Metal–Organic Framework (MOF) Composites

The extensive use of fossil fuels has led to excessive emissions of $CO_2$, which has caused serious ecological hazards. Visible light photochemical conversion technology is similar to photosynthesis, which has the advantages of having a clean energy source and doing no harm to the environment. Using solar energy to reduce the $CO_2$ levels in high value-added organic chemicals is an important means of solving the energy crisis and environmental problems. Compared with gasoline, methanol is more eco-friendly, has a higher energy density and octane number, and is a promising product for $CO_2$ photoreduction. A MOF is a kind of crystal material composed of metal ions/clusters and organic ligands. They are widely used in gas adsorption/separation, sensors, drug delivery, heterogeneous catalysis, and other fields.

### 3.3.3. Comprehensive Utilization of Solid Waste

In 2020, China will produce more than 10 billion tons of solid waste, including 6 billion tons of organic solid waste represented by kitchen waste, straw, livestock, and poultry manure and 2.5 billion tons of industrial solid waste represented by tailings, fly ash, coal gangue, smelting waste residue, slag, and desulfurized gypsum. The annual output of renewable resources, represented by waste components, plastics, and metals, exceeds 300 million tons, and the output of industrial hazardous wastes with high pollution risks is about 40 million tons.

At present, there are still 6 billion tons of bulk solid wastes in China, which lack economic and reasonable methods of disposal and utilization and are still mainly landfilled after disposal. They are a low standard of resource utilization and occupy a large amount of land for stockpiling. In the process of the treatment and disposal of solid waste, greenhouse gases such as carbon dioxide, methane, and nitrous oxide are emitted. The contribution rate of solid waste to greenhouse gases in different countries is between 2% and 3.5%. Moreover, the complex toxic substances contained in it can easily cause the pollution of water, soil, air, and other media. In order to achieve the goal of a carbon peak and carbon neutrality, the international community is not only paying attention to the carbon emissions of industry and transportation, which have high carbon intensity, but also carbon emissions in the context of solid waste, which can be reduced in various ways. With considerable effort, the EU has reduced the total greenhouse gas emissions from waste from 240 million tons in 1990 to 135 million tons in 2019, which represents a decrease of 43.8%. This reduction in solid waste treatment accounts for 86% of the total emissions reductions in the field of waste within the EU.

It should be emphasized that China's emerging industries are booming, and the production of emerging solid waste is also undergoing explosive growth. By 2035, China will produce more than one million tons of waste in the form of photovoltaic modules, wind turbine blades, and power batteries alone every year. However, this solid waste contains a variety of rare and precious metal resources, such as silver, tellurium, indium, gallium, etc. If it is not properly handled, this waste could seriously pollute water, air, soil, and other environmental media. It can be seen that solid waste treatment and disposal in

China must strictly follow the concept of low-carbon ecology, strengthen the whole process control technology route of "source reduction, process recycling, and end harmlessness", and give full play to the synergistic effect of the comprehensive utilization of solid waste in the realization of pollution reduction and carbon reduction goals [12–17,19].

### 3.4. Karst Carbon Sinks

3.4.1. Theoretical Research

The karst carbon cycle refers to the process by which carbonate weathering and dissolution consume atmospheric or soil $CO_2$ to form inorganic carbon and in which the carbon forms are transformed and transported by physical, chemical, and biological processes in karst basins. On this basis, the term karst carbon sink refers to the dissolved inorganic carbon (DIC), dissolved organic carbon (DOC), and inert organic carbon (RDOC) in the karst basin, which will consume atmospheric or soil $CO_2$ through the dissolution of carbonate rocks. These processes and fluxes are retained or fixed in the hydrosphere and biosphere. Karst carbon sinks are considered important carbon sinks. At present, there are two prominent barriers to the effective use of karst carbon sinks. First, there is a lack of accurate and systematic assessments of karst carbon sinks in karst areas. Second, it is not known whether the karst carbon sink can be increased through human intervention. The first problem that needs to be addressed is how to effectively calculate the rate and flux of organic and inorganic carbon sinks in the process of rocky desertification restoration and how to quantify the contribution rate of vegetation organic and karst inorganic carbon sinks to carbon neutralization in the process of ecological restoration. It is necessary to develop the calculation method of karst carbon sink according to the karst geological background and ecological conditions, to determine the relevant parameters of the inorganic-organic carbon cycle, and study the mechanisms and processes involved. The second problem is that the main factors affecting the dissolution of carbonate rocks are $CO_2$ and water. Afforestation in karst areas should not only emphasize the improvement of vegetation coverage but should also pursue the overall improvement of the ecosystem stability and service functions. High photosynthesis and high carbon sequestration capacity restoration, vegetation selection, increased vegetation coverage, positive succession of vegetation, and soil improvement can increase $CO_2$ concentration and cycle speed. The introduction of non-karst water to irrigate vegetation and crops in karst areas can increase the effect of the karst carbon sink.

Around 70–80% of the carbon cycle in karst occurs in the superficial karst zone, and only a small portion occurs in underground rivers and caves. In the migration process of karst water rich in $HCO_3^-$, a small amount of $HCO_3^-$ is converted into $CO_2$ to escape to the cave air. More $HCO_3^-$ then flows out of the surface in the form of springs and underground rivers with the flow of groundwater and becomes surface rivers. These karst waters with a high concentration of inorganic carbon content stimulate aquatic plants to carry out photosynthesis so that part of the inorganic carbon is converted into organic carbon.

3.4.2. Methods for Increasing the Carbon Sink Effect

According to the karst carbon cycle, the karst carbon sink flux and its stability can be artificially improved in the four following technical ways:

(1) Vegetation ecological restoration. In view of the fact that the driving force of plant photosynthesis and carbonate weathering and dissolution is $CO_2 + H_2O$, vegetation ecological restoration can not only increase the carbon sink flux of surface organisms but can also increase the carbon sink flux of underground karst. From shrubs to secondary forest land and then to virgin forest land, the carbon sink flux produced by karstification can increase by 2 to 8 times.

(2) Soil improvement. The carbon in a karst carbon sink mainly comes from soil $CO_2$, and the concentration of soil $CO_2$ is 1 to 2 orders of magnitude higher than that of the atmosphere. The effect of the karst carbon sink can be strengthened by improving the soil and increasing the soil $CO_2$ cycle.

(3) Enhance the photosynthesis of aquatic plants. The high concentration of $HCO^{3-}$ in the water body of a karst area is very unstable. When the hydrological conditions change, it often transforms into $CO_2$ and escapes to the atmosphere. If the photosynthesis of aquatic plants consumes some of the $HCO^{3-}$ and reduces its concentration, the inorganic carbon will be converted into organic carbon, which can greatly improve the stability of the carbon migration process in karst water [14–17,19].

## 4. Case Analysis of Green and Carbon-Neutral Coordinated Development in Guizhou

In the period of the transformation of the traditional growth model of "mass production, mass consumption, and mass waste", if Guizhou province does not change the traditional resource development economic model, the resource input and pollution emissions required in the future will double, which may cause greater environmental disasters, hamper economic growth, interrupt development, and make it difficult to achieve a historic leap. Therefore, in order to save energy and achieve clean and sustainable economic development, we must rely on the circular economy, which can adjust the production mode of the national economy in Guizhou province and solve the problem of energy saving and emission reduction. In the process of green carbon neutralization and coordinated and innovative development, Guizhou must adhere to the planning guidance, implement the concept of green development, adhere to its ecological priority, deepen the adjustment of its industrial structure, adhere to the demonstration drive, build a green manufacturing system, adhere to its low-carbon transformation plan, vigorously improve the level of energy efficiency, adhere to its circular development plan, and improve the efficiency of the comprehensive utilization of resources.

### 4.1. Guizhou Phosphate Chemical Group

The annual phosphate mining capacity of Guizhou Phosphate Chemical Group exceeds 17 million tons. Its annual phosphoric acid production capacity exceeds 4 million tons. Its annual high concentration phosphate compound fertilizer and new fertilizer production capacity exceeds 8 million tons. Its annual synthetic ammonia production capacity exceeds 1.4 million tons. Its annual wet purification phosphoric acid production capacity reaches 1 million tons. Its annual phosphate production capacity reaches 500,000 tons. Its annual synthetic ammonia (including alcohol) production capacity is 1.4 million tons. Its annual anhydrous hydrogen fluoride production capacity is 90,000 tons.

Guizhou Phosphate Chemical Group has the only State Key Laboratory for the Efficient Utilization of Medium- and Low-Grade Phosphate Ore and Its Associated Resources in the industry, as well as technological innovation platforms such as post-doctoral research workstations and state-level enterprise technology centers. At the same time, it has also cooperated with more than 30 well-known universities and research institutes at home and abroad to develop dozens of international and domestic leading core technologies covering the whole phosphorus chemical industry chain, such as low-grade phosphate ore mining, phosphogypsum underground filling, wet-process phosphoric acid purification, the comprehensive utilization of fluorine and iodine resources associated with phosphate ore, and environmentally protective disposal of waste, all of which have achieved scale and industrialization (Figure 4).

Guizhou Phosphate Chemical Group improves the comprehensive utilization efficiency of resources through industrial chain synergy and technological innovation and explores new, green, low-carbon, high-quality methods of sustainable development in industry. From the perspective of energy and resources, enterprises should try their best to realize comprehensive utilization and recycling. For example, the waste heat generated in the production process can be converted into energy reuse, and carbon dioxide waste gas can be converted into liquid carbon dioxide and dry ice, which can help it to realize its commercial value while reducing carbon emissions. In addition, phosphogypsum, a by-product of the phosphorus chemical industry, is harmful to the environment. Through scientific and technological innovation, Guizhou Phosphate Chemical Group has made

great efforts to solve the problem of the comprehensive utilization of phosphogypsum. The carbon emissions of phosphogypsum-developed building materials is only one seventh that of traditional cement building materials. This measure provides a systematic solution to environmental protection and carbon reduction in building materials. At the same time, Guizhou Phosphate Chemical Group also promotes low-carbon transformation and green development by actively developing new energy battery cathode materials and electrolytes.

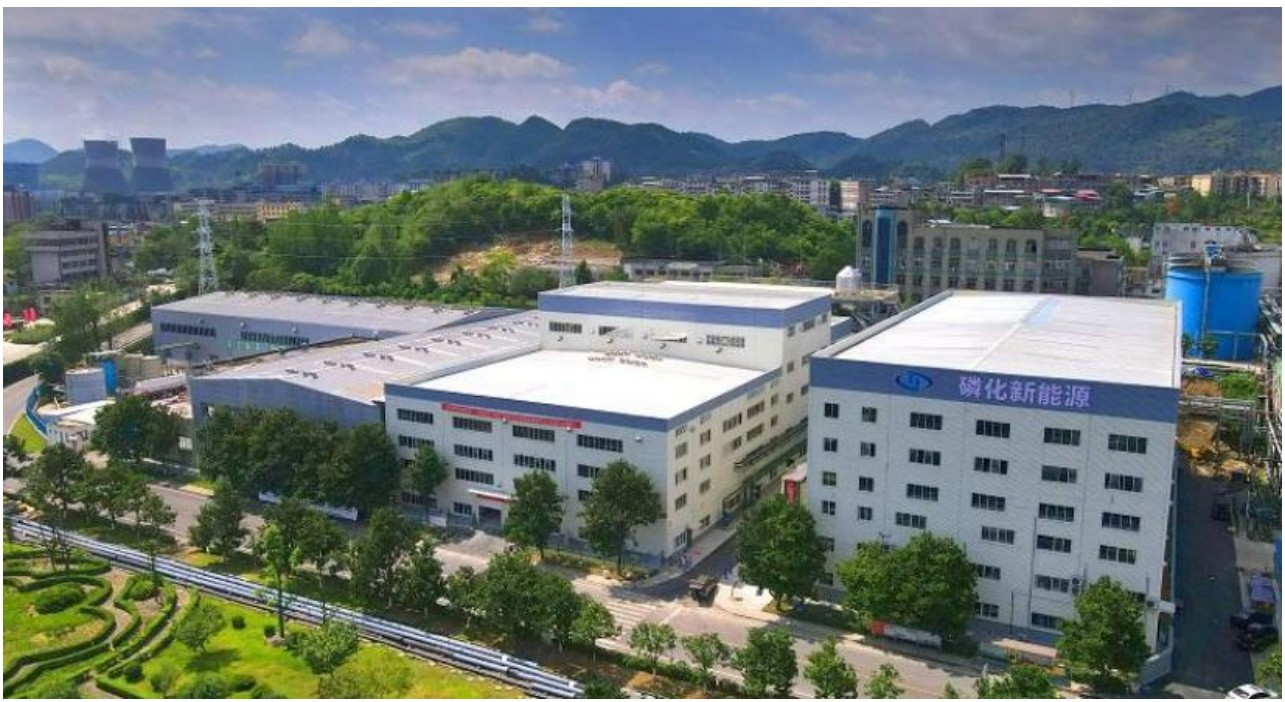

**Figure 4.** The new energy plant of the Guizhou Phosphate Chemical Group.

*4.2. Green Economy Development of Coal Resources in Liupanshui City*

Liupanshui is an industrial city that relies mainly on energy and raw materials. Its industries are mainly concentrated in coal, metallurgy, electric power, and building materials. At the present stage, the city's development mode is in a critical period of transformation from extensive to intensive, and the "three wastes" are still in a high-emission period. In order to practically implement a scientific outlook in terms of development and to minimize resource waste and environmental pollution, we should adhere to the goal of building a resource-saving and environmentally friendly society, focus on energy and raw materials, and take the development of the circular economy as an important measure to adjust the economic structure, change the mode of development, and achieve the goal of energy conservation and emission reduction in accordance with the requirements of "reduction, reuse, and recycling". All coal-washing water realizes closed circulation, all thermal power plants have installed desulfurization facilities, all coking enterprises have built chemical recovery systems, and all cement enterprises have been upgraded from the wet production mode to the new dry production mode. Large, state-owned enterprises are encouraged to actively develop circular economies and are supported when doing so. Circular development modes, such as coal washing, slime drying and utilization, mine water treatment and recycling, coke oven gas co-combustion for power generation, waste heat and pressure power generation, new wall materials made of pulverized coal, coal mine gas power generation and concentrated canned civil use, and coal liver stone power generation have been implemented. Coal, electric power, metallurgy, chemical, building materials, and other industrial clusters in the city have gradually developed, and resource advantages are being converted to economic advantages (Figure 5).

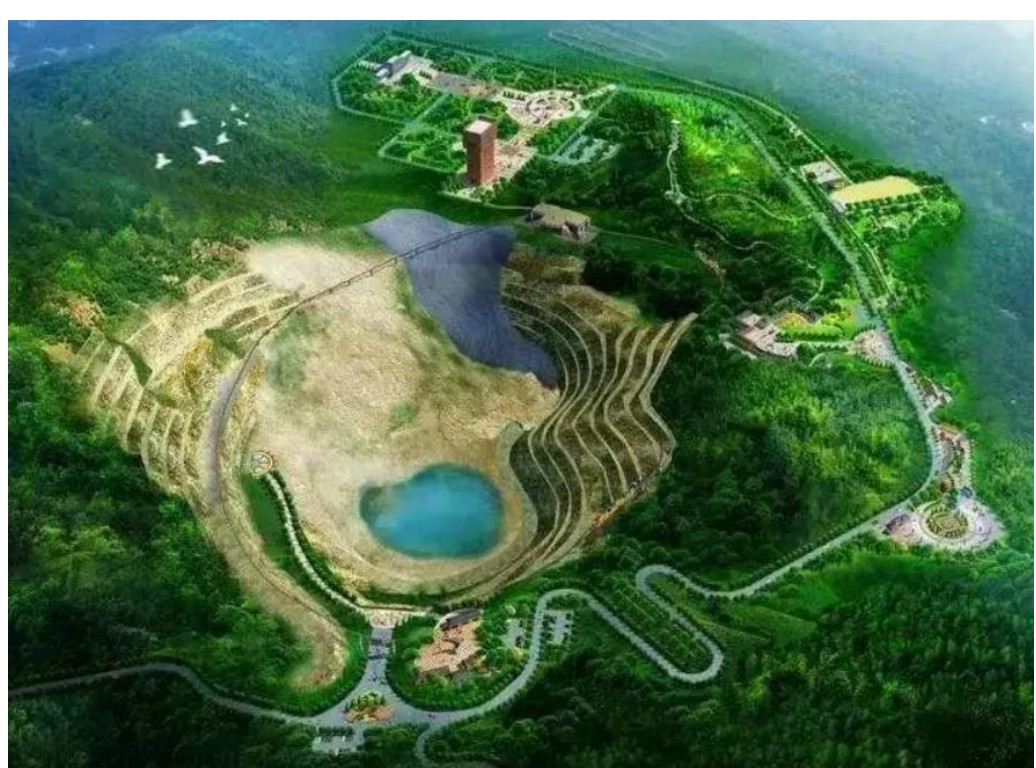

**Figure 5.** Mine ecological restoration in Liupanshui City.

*4.3. Campus Carbon Emission Management at Guizhou Institute of Technology*

The Guizhou Institute of Technology is located in Yunyan District, Guiyang City, Guizhou province and covers an area of 866.7 mu. It has a total of more than 14,000 faculty members and students. According to the statistics of the main types of carbon emissions at the Guizhou Institute of Technology, the total amount of carbon emissions per year is 15,227 t The carbon emissions of human metabolism account for 4599 t, the carbon emissions of direct electricity consumption account for 3365.85 t, the carbon emissions from oil and gas were 1208.66 t, the carbon emissions from water consumption were 4225.38 t, and the carbon emissions produced by other daily activities were 1828.9 t, accounting for 30%, 22%, 8%, 28%, and 12% of the total, respectively (Table 1).

**Table 1.** Statistical table of carbon emission estimation at Guizhou Institute of Technology.

| No. | Item (unit) | b (Carbon Footprint Factor) | *n* (Quantity) | Carbon Emission (t/a) |
|---|---|---|---|---|
| 1 | Personnel Metabolism | 0.9 kg/P/d | 14,000 persons | 4599 |
| 2 | Electricity | 0.9344 kg $CO_2$/kWh | 3,602,151 kWh | 3365.85 |
| 3 | Natural gas | 0.4483 kg $CO_2$/m$^3$ | 89,158 m$^3$ | 39.97 |
| 4 | Diesel | 3.159 kg/L | 11,202.4 L | 35.39 |
| 5 | Gasoline | 2.985 kg/L | 379,668.7 L | 1133.3 |
| 6 | Water | 0.9344 kg $CO_2$/kWh | 4,522,028.06 kWh | 4225.38 |

The school has made a long-term commitment to conduct campus energy-saving. A self-service recycling station has been set up at the school where students can recycle plastic bottles, unused books, and other items. For this, they obtain a certain amount of money. This helps improve the enthusiasm of personnel and cultivates energy-saving awareness. It calls on teachers and students to save electricity. The current teaching building uses a master control switch, which allows the staff to manage the lights in the classroom but also causes a problem that the lights in classrooms are not turned off. At present, the solution is to remind students that they should remember to turn off the lights when entering and leaving a classroom. They also regularly arrange for personnel to patrol the

corridors of the teaching building and turn off the lights when there are no students in a classroom. In addition, the technical transformation of regional lighting facilities such as roads and green belts in colleges and universities is advocated to achieve green lighting. It is strictly forbidden for students to use high-power electrical appliances in dormitories, which not only ensures the safety of students in dormitories but also greatly reduces the electricity consumption of student apartments. Starting from strengthening the regular inspection and maintenance of water equipment in order to avoid the phenomenon of dripping and leaking water, we should cultivate the habit of conscious water-saving of school personnel, turn off the faucet at will, put an end to the phenomenon of running water for a long time, strictly enforce the time of water supply in student dormitories, supply water during a specified period of time, and reduce the supply of water. In addition, sprinkler irrigation and drip irrigation should be used for campus greening to avoid the loss of most water resources. We should strengthen the management of water use in school projects so as not to discharge water wastefully. Additionally, materials are saved. Online office mode can be promoted, which can not only reduce the consumption of office equipment but also achieve document networking, and personnel should be arranged to regularly manage and supervise the use of paper in school public printers. Strengthen comprehensive application. Purchasing instruments with less energy consumption not only meets the needs of teachers and students but also reduces carbon emissions. Disposable bowls and chopsticks and disposable food bags should not be used or should be used less. Energy-saving operations in campus buildings is the key goal of low-carbon campus construction. A building monomer can be regarded as an organic unit of a campus, and its planning, design, construction, use, maintenance, renewal, and other links directly affect the energy demand for heating, air conditioning, lighting, hot water, and so on. Therefore, the campus carbon emission-reduction strategy adopts a systematic view, from the whole to the local layer by layer (Figure 6).

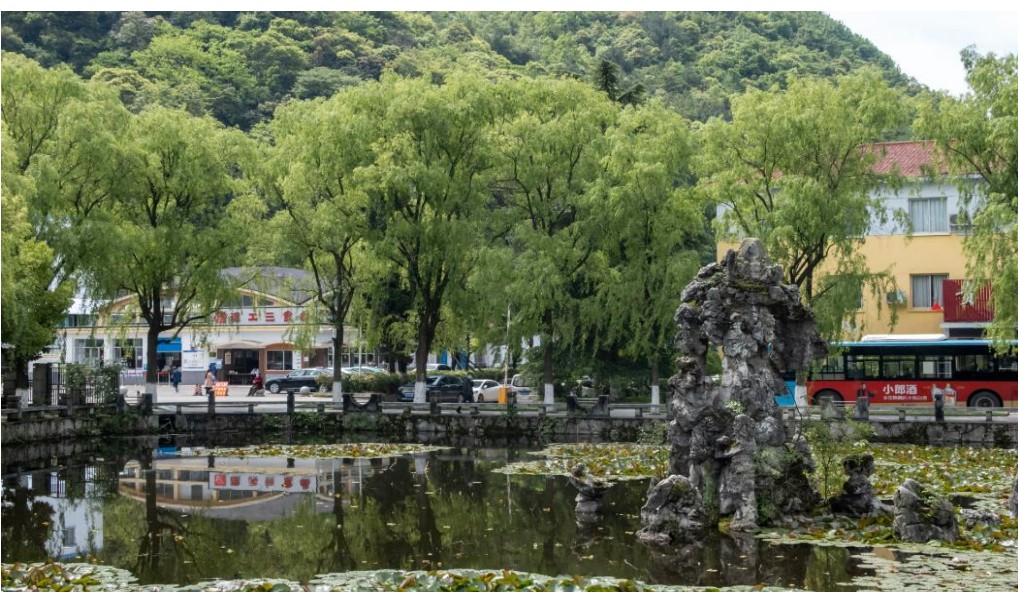

**Figure 6.** Green planting at Guizhou Institute of Technology.

## 5. Discussion

Through the current research and the systematic summary and analysis of the existing innovative technologies, we find that Guizhou is facing many difficulties and problems in the reality of green development and carbon neutrality. In recent years, many promising advancements have been made. In order to explore the path of carbon emission reduction adapted to national conditions and achieve China's carbon neutrality goal, the first task is to find out the real-life dilemmas involved in achieving the carbon neutrality goal. China's

carbon neutrality has developed good momentum, but it still lags behind others in energy transformation, systems, and marketization [14–17,19].

### 5.1. Energy Transformation Is Difficult

According to the World Energy Statistics Yearbook 2021 released by BP, China's total carbon emissions reached 9.899 billion tons in 2020, accounting for 30.7% of the world's total. In terms of energy structure, according to the data of the National Bureau of Statistics, the consumption of clean energy such as natural gas, hydropower, nuclear power, wind power, and solar power accounted for 25% of total energy consumption in 2021. Coal consumption accounts for 56% of the total energy consumption, and "one coal dominates" and seriously restricts the process of carbon emission reduction in China. In terms of energy intensity, although China's energy intensity has gradually declined in the past decade—its carbon emission intensity in 2019 was reduced by 48.1% compared with 2005, and the target of reducing the carbon emission intensity by 40–45% proposed in 2015 has been achieved ahead of schedule—it is still higher than the world average level. It can be seen that although China's carbon emission intensity is continuing to decline and basically curbs the trend of the accelerated growth of carbon emissions, such carbon emission characteristics and energy consumption structures make it difficult to reduce the dependence on coal in a short time. Thus, China's energy transformation still faces severe challenges, and China's road to carbon neutrality still faces tremendous pressure.

On the other hand, most developed countries have basically reached their carbon peaks in the early 21st century. Although some European countries are affected by the oil crisis, environmental policies, and the extensive use of natural gas, things reach a natural peak when economic development has entered a mature stage and the per capita GDP has reached about 25,000–40,000 USD. Europe has experienced a transitional period of 50–60 years, which is a natural process of technological and economic development. By contrast, China is at the stage of industrialization and urbanization, and the task of economic development is arduous, so it is necessary to maintain the sustained growth of energy consumption and carbon emissions. The transition time from carbon peak to carbon neutrality is only 30 years, and the per capita GDP is only about 12,000 US dollars, which is significantly lower than the peak level of many developed countries. This means that there is no buffer period for China's carbon peak, which is totally different from the internal and external environment faced by developed countries, and thus the task of reducing emissions in China is arduous [18,20–22].

### 5.2. Carbon Neutral Technology Needs to Be Improved Urgently

China attaches great importance to the development of zero-carbon energy technology. According to statistics from the National Energy Administration, by the end of 2021, the installed capacity of renewable energy in China has reached 1.063 billion kilowatts, to which hydropower, wind power, and solar power make great contributions. According to data from the China Nuclear Energy Industry Association, by the end of 2021, 53 nuclear power units were in operation in China, and the installed capacity was 54,646.95 MW, ranking third after the United States and France, with nuclear power accounting for 5.05% of the total power generation, meaning that the role of clean energy in the power supply has been further enhanced. However, China is still in the stage of industrialization, and the demand for electricity will increase rigidly, so it is difficult to make up for the energy consumption of thermal power via renewable energy generation alone. In addition, the high cost of clean energy storage makes it difficult to alleviate the "intermittent" curse. Compared with thermal power, the intermittency and stability of wind and solar energy lead to a high cost of energy storage, which requires the grid to have intelligent peak shaving capacity. Considering the GDP factor, some local governments are not enthusiastic about the acceptance of clean energy, which leads to serious wind and light abandonment. According to statistics, the total rate of wind abandonment in Xinjiang, Gansu, and Inner Mongolia is 30%.

In terms of sink increasing technology, China is rich in forestry and marine resources, and forests, grasslands, and oceans have the potential for "absolute carbon sequestration" emission reduction and are known as "green gold" and "blue treasure". The development of forest carbon sequestration is relatively mature. The pilot project of forestry carbon sequestration started in 2004. By 2021, China's forest coverage rate will reach 23.04%, the forest area will be 220 million hectares, and the carbon sequestration capacity and potential will be very significant. At the same time, combined with the characteristic ecological concept of "green water and green hills are Jinshan and Yinshan", new formats such as "ecological banks" are constantly emerging to promote the coordinated development of the ecological economy. However, the proportion of forestry carbon sequestration projects in China's CCER projects is small, accounting for only 1% of the average annual emission reduction of 670 million tons of CCER projects in China, and forestry carbon sequestration is facing market failure.

The reason is that forestry carbon sequestration projects are generally small in scale, and the cost of daily management and monitoring is very high; the estimation methods of carbon sequestration in grassland, wetland, and farmland ecosystems are different, which brings great uncertainty to the estimation results of carbon sequestration, and there is a lack of different methods of ecosystem carbon sequestration technologies and models, as well as a unified national standard system for carbon sequestration measurement, assessment, and technology (Figure 7).

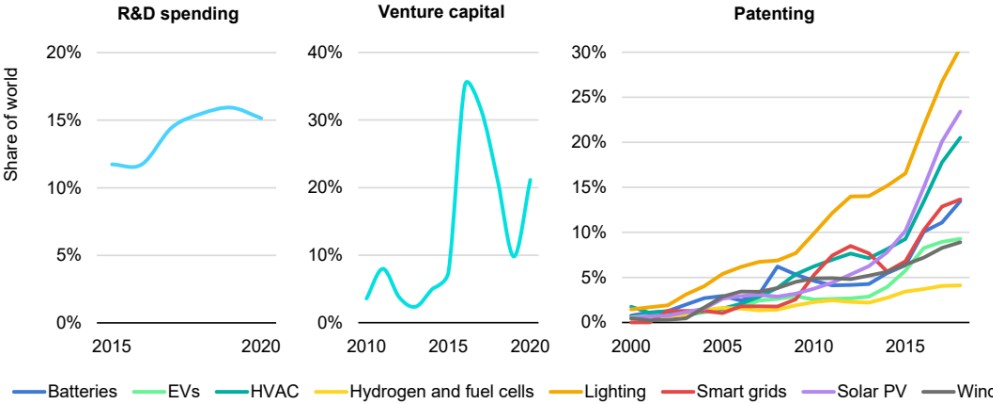

**Figure 7.** China's share of global public spending on low-carbon energy R&D, venture capital, and patenting [15].

*5.3. The Market Incentive Mechanism Is Not Effective*

In addition to encouraging progress in clean energy technology, the market incentive mechanism for carbon emission reduction is also one of the core tools for implementing a vision of carbon neutrality. At present, the market incentive mechanism for reducing carbon emissions is mainly divided into two categories: one is the price-based emission-reduction tool (carbon tax), and the other is the quantitative emission-reduction tool (carbon emissions trading). At present, carbon tax and carbon trading instruments, as the most important market incentive mechanisms, are widely used in developed countries such as the European Union, the United Kingdom, and the United States. China launched carbon trading pilot projects in seven provinces and cities in 2011 and began trading in the second half of 2013. The total population covered by the pilot projects has reached 260 million, the energy consumption was 830 million tons of standard coal, and the GDP reached 14 trillion CNY, accounting for 19%, 27%, and 23% of the country, respectively. In 2016, a new carbon trading market was added in Fujian Province. As of December 2020, the total turnover of carbon market quotas in eight provinces and cities in China was 445 million tons, with a turnover of 104. The total turnover of CCER was 268 million tons. In 2021, the national unified carbon market was launched, covering the power generation industry and including 2162 power generation enterprises as key emission units. As of

21 July 2022, the cumulative turnover of carbon quotas in the national carbon market was 194 million tons, with a cumulative turnover of 8.505 billion CNY. Generally speaking, China's carbon trading pilot projects have certain reference value in terms of their total scale, but the emission-reduction effect of the carbon market still needs to be improved, and there are some problems such as the imperfect policy framework of the carbon market, incomplete financial attributes, and inactive trading. Firstly, China's carbon market is not active. The phenomenon of "heavy performance and light trading" is more common, and the carbon price is low, which fails to give full play to the price discovery function of the carbon market. The long-term active carbon market is the basis for ensuring sufficient liquidity and finding a reasonable carbon price, but the price of carbon emission rights in China's pilot areas varies significantly, with an average of less than 30 CNY/ton, and there is no effective price discovery function to solve the balance between economic growth and carbon emissions. The reason is that China has not built the financial function into the carbon market, resulting in a lack of hard constraints on emission reduction and a lack of internal motivation for local governments and market participants to achieve carbon emission reduction. Secondly, carbon finance innovation is insufficient, and the participants and products are singular. In terms of product structure, the main body of the carbon market is still spot trading, and China's certified voluntary emission reduction is low. Although some carbon trading pilot projects open carbon funds and carbon emission rights pledge financing and other carbon financial innovative products, they are still in a sporadic pilot state with unbalanced regional development and a lack of systematic and perfect carbon financial innovative products. The current product structure still cannot meet the carbon asset management needs of emission control enterprises, let alone radiate and serve carbon trading in other regions. In terms of trading subjects, most of the participants are emission control enterprises, few pilot projects allow individuals to participate, the market activity is not high, the supply side of carbon trading and market demand forces are unbalanced, and the contradiction between supply and demand is bound to suppress the activity of carbon trading. Carbon tax and carbon trading are essentially the pricing of carbon emissions, and in theory, the policy effect is the same, which can help to achieve Pareto optimality. Therefore, in the long run, we need to speed up the landing of carbon tax. However, when and how to levy carbon taxes to help achieve the goal of carbon neutrality fairly and effectively is one of the practical dilemmas facing China [22–28].

## 6. Suggestions

The "14th Five-Year Plan" period is the key five years for Guizhou province to implement new development concepts, to integrate into the new development pattern, and to promote high-quality development based on a new development stage. The carbon peak and carbon neutralization are effective ways for Guizhou province to take the overall situation into account to help achieve high-quality development. This is the only way for Guizhou province to accelerate new industrialization, new urbanization, agricultural modernization, and tourism industrialization. Achieving the goal of carbon peak and carbon neutralization is conducive to the formation of an inverted mechanism to promote the fundamental transformation of the industrial structure and energy structure, and the thorough transformation of the production mode and lifestyle from the source; to promote the three strategic actions of rural revitalization, big data, and big ecology throughout the province; and to speed up the construction of a high-quality industrial system, a high-standard urban system, and a high-quality rural construction system. As an important energy base and resource processing base in China, the energy sector has always been the main source of carbon emissions in Guizhou province, and the energy sector's carbon emissions are close to being half of the total carbon emissions in the province. Whether the energy sector can successfully reach the carbon peak in time is directly related to the overall situation of the carbon peak and carbon neutralization in Guizhou province. Taking the opportunity to establish a new national comprehensive energy strategic base, we should make concerted efforts involving the intelligent green development of the basic energy

industry and the clean and efficient development of the power industry and continue to promote the green upgrading of energy infrastructure, the low-carbon transformation of energy production, the electrification of energy consumption terminals, and the deepening of the reform of energy systems and mechanisms. Efforts should be made to build a green, low-carbon, safe, and efficient modern energy system. Through the transformation of stock energy to enhance carbon reduction and make energy clean, efficient, and low-carbon, we will make great contributions to achieving the early carbon peak of the whole province and even the whole country.

*6.1. Coordinating the Green and Low-Carbon Transformation of the Energy Sector*

(1) we need to coordinate the relationship between transformation and security. The two goals of carbon reduction and supply guarantee must be achieved simultaneously, the transformation must be safe, and safety is the bottom line that must be adhered to. For example, Guizhou's resource endowment, energy structure characteristics, and development trend determine that Guizhou needs coal and coal power as the "ballast stone" to ensure energy security during the "14th Five-Year Plan" period, and the role of coal and coal power should be strengthened. It is estimated that in order to ensure the safe operation of energy and power in Guizhou province, it needs to increase the installed capacity of coal-fired power by 7.5–11.5 million kilowatts during the "14th Five-Year Plan" period. In the process of reaching a carbon peak in the power industry, the premise should be to ensure the safe supply of energy and power.

(2) we need to coordinate the relationship between traditional energy and new energy. We should vigorously promote safe, green, and intelligent coal mining and the clean and efficient utilization of coal, reduce the energy consumption level of active coal power, vigorously develop new energy sources such as photovoltaic power generation and wind power, continuously improve the level of clean energy consumption, and continuously adjust and optimize the energy structure. At the same time, in order to ensure the absorption and delivery of large-scale new energy, the peak-shaving power supply of new energy such as pumped storage, gas, and electricity should be developed to ensure that the system has sufficient peak-shaving capacity and the installed capacity required for the stable operation of the system.

(3) we should coordinate the relationship between carbon reduction in terms of stock and low-carbon energy in increments. In the process of reaching the carbon peak, the balance between stock and increment, as well as between long-term goals and short-term tasks, should be taken into account. We should not only vigorously promote the upgrading and transformation of coal-fired power in terms of stock and reduce coal consumption but should also rationally plan and develop coal-fired units with high parameters, high efficiency, and ultra-low emissions. We should reasonably arrange the scale of annual unit transformation and the scale of new production.

*6.2. Improve the Production System of Green Low-Carbon Cycle Development*

(1) we need to promote the green upgrading of industry. We will resolutely curb the blind development of "two high" projects, accelerate the implementation of the intelligent transformation of equipment and iterative upgrading of processes involving the iron and steel, chemical, non-ferrous, and building material industries, accelerate the application of integration technology, and reduce material consumption and pollutant emissions from the source. We will promote the construction of demonstration bases (enterprises) for the comprehensive utilization of bulk solid waste and bases for the comprehensive utilization of industrial resources.

(2) we need to accelerate the green development of agriculture. For example, Guizhou is promoting the processing industry of agricultural products and brand cultivation and has designated 12 characteristic and advantageous agricultural industries, including fine tea, edible fungi, pepper, traditional Chinese medicine, and Rosa roxburghii Tratt.

(3) we need to improve the level of green development of the service industry. We need to promote the green upgrading of commercial and trade enterprises and cultivate a number of green circulation subjects. We need to develop the shared economy of the life services industry in an orderly manner and improve the utilization rate of idle resources.

(4) we need to expand the green environmental protection industry. We will accelerate the development of green environmental protection industries such as aerospace, high-end equipment manufacturing, new energy vehicles, biomedicine, and new materials. We should also promote the construction of a national green industry demonstration base in the Qingzhen Economic Development Zone.

(5) we need to enhance the recycling level of industrial parks and industrial clusters. We will promote the comprehensive implementation of recycling transformation in industrial parks at the provincial level and promote the co-construction and sharing of public facilities, cascade energy utilization, resource recycling, and pollution, as well as the centralized and safe disposal of pollutants, etc.

*6.3. Improve the Consumption System of Green Low-Carbon Cycle Development*

(1) We need to promote the consumption of green products. We will intensify government green procurement, expand the scope of green product procurement, and promote the incorporation of green procurement into the enterprise procurement systems of state-owned enterprises. We will strengthen guidance for enterprises and residents that encourages them to purchase green products and will encourage local governments to promote green consumption by means of subsidies and bonus points.

(2) We need to advocate a green and low-carbon lifestyle. We should guide catering service providers to strengthen their management in terms of their storage management, processing and production, dining services, and other links so as to avoid waste in the process of catering operation. We should urge catering service providers to guide consumers to order and take meals in appropriate amounts, reduce the use of disposable tableware, and resolutely stop waste in catering. We should strengthen the chain management of plastic pollution. We will carry out activities to create green lifestyles, green ecological families, green schools, green communities, and conservation-oriented organs.

*6.4. Consolidate and Enhance Carbon Sink Capacity*

(1) We need to consolidate and enhance the carbon sink of the forestry ecosystem. This involves carrying out land greening and beautification actions, consolidating the achievements of returning farmland to forests and grasslands, strengthening the classified protection and construction of natural forests, public welfare forests, shelter forests, reserve forests, and carbon sink forests, strengthening forest management and tree species structure adjustment, promoting the transformation of low-yield and low-efficiency forests and the restoration of degraded forests, increasing forest and grassland fire prevention and forest pest control capacity, and improving forest and grassland disaster prevention and mitigation capabilities. We should also promote the ecological restoration of historical mines in key areas. We will continue to promote forestry carbon sequestration and single-plant carbon sequestration projects and establish and improve a system that can reflect the carbon exchange rate.

Value compensation mechanism for ecological protection.

(2) We need to steadily improve the carbon sink capabilities of farmland, grassland, and wetland. We should carry out actions to reduce emissions and to sequester carbon in agriculture and rural areas, and to vigorously develop green, low-carbon, and circular agriculture. We should apply agricultural technology to increase carbon dioxide sequestration and to explore and popularize the technology of carbon dioxide gas fertilizers. We should speed up the completion of farmland infrastructure, continuously improve the quality of cultivated land, and increase soil organic carbon reserves. We should strengthen the resource utilization of crop straw and livestock and poultry manure. We should rationally protect karst topography and landforms and speed up the development and utilization

of karst carbon sinks. We should strengthen grassland protection and restoration; carry out artificial grass planting, grassland improvement, and fencing construction; and adopt measures such as grassland improvement, artificial grass planting, reseeding, fertilization, and fencing to restore grassland vegetation in rocky desertification areas so as to gradually improve grassland productivity and grassland comprehensive vegetation coverage. We should accelerate the protection and restoration of ecological wetlands, improve the system of wetland protection and management, and strengthen the dynamic monitoring of wetland resources [29–34].

**Author Contributions:** Conceptualization, W.Y.; methodology, W.Y.; software, Z.M.; formal analysis, W.Y.; investigation, Z.M. and W.Y.; data curation, J.Y.; writing—original draft preparation, W.Y.; writing—review and editing, M.Y. and W.Y.; visualization, J.Y.; supervision, J.Y. All authors have read and agreed to the published version of the manuscript.

**Funding:** This research was supported by the Science and Technology Support Plan of Guizhou Province (Qian (2017)1410); the High-Level Talent Introduction Program for the Guizhou Institute of Technology (0203001018040). Guizhou Institute of Technology 2022 Rural Revitalization Soft Science Project 2022xczx10; Guizhou Provincial Science and Technology Projects [QKHJC-ZK(2022)-General186]; Projects of National Natural Science Foundation of China [No. 41602275].

**Institutional Review Board Statement:** Not applicable.

**Informed Consent Statement:** Not applicable.

**Data Availability Statement:** Not applicable.

**Conflicts of Interest:** The authors declare no conflict of interest.

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
