# Peer review of "Innovation Strategy for Green Development and Carbon Neutralization in Guizhou—An Overview"

_sustainability, doi:10.3390/su142114377_

Round 1

Reviewer 1 Report

Dear Aythors

I am sorry but your article requires a thorough reconstruction. Here are my comments: 

I propose the title: "Innovation Strategy of Green Development and Carbon Neutralization in Guizhou — An Overview"

2. In the chapter Introduction, the Author exaggerated by saying: "These commitments and goals not only demonstrate China's responsibility to actively respond to global climate change, but also demonstrate China's determination and courage to promote a comprehensive green transformation in China and accelerate the formation of a clean, efficient, green and safe modern governance system. " - It does not require courage and determination, but common sense in making decisions and the integrity of scientific research on the basis of which decisions are made. The authors' statements are of a journalistic nature.

3. In the Introduction section of the last paragraph, briefly and succinctly inform the reader what the purpose of this work is and why it is important to achieve this goal.

4. Chapter 2 adds nothing significant to the topic discussed. This is journalistic content that does not indicate the scientific value of this work. I propose to delete it or radically shorten it, pointing only to the significant development trends that the Author intends to discuss in the review of the thematic literature.

5. Chapter 2 should be drafted as comments on the thematic publications cited. There are no publications relevant to the topic in this chapter.

6. Does Figure 4 show the authors' research or is it based on data from other publications?

7. "Case Analysis of Green and Carbon Neutral Coordinated Development In GuiZhou" - the title of chapter 4 is misunderstood - what the Authors understand by the term natural development.

8. Chapters 3 and 4 too extensive descriptively of a journalistic nature. It should be specified and made scientific in nature.

9. In Chapter 5, conclusions important for the subject should be formulated.

Taking into account the above suggestions, the article should be thoroughly prepared in accordance with the subject matter. In this form, it is not suitable for publication in a scientific journal.

Greetings

Rewiever

Author Response

Dear reviewer:

Thank you very much for your work.We have learned much from your and other two reviewers’ comments,which are fair,encouraging and constructive.After carefully studying the comments and your advice, we have made corresponding changes.

  1. According to the suggestions of the reviewers, the title is optimized.

The new title is:Innovation Strategy of Green Development and Carbon Neutralization in Guizhou — An Overview

  1. The authors' statements are of a journalistic nature.

This problem does exist. In order to highlight the policy, there is news administration in the introduction part. We have deleted this part and simplified it.

  1. In the Introduction section of the last paragraph, briefly and succinctly inform the reader what the purpose of this work is and why it is important to achieve this goal.

We rewrite this part, highlighting the research content, research ideas and purpose in the last paragraph, and provide suggestions for green carbon neutralization in Guizhou on the basis of summarizing the existing situation and technology.

In order to tackle the climate change crisis and achieve the temperature control goals of the Paris Agreement, China has made a solemn commitment to "peak carbon by 2030 and carbon neutral by 2060", raising the proportion of non-fossil energy from 20% to 25%, and increasing forest stock from 4.5 billion cubic meters to 6 billion cubic meters. Achieving the goal of carbon peak and carbon neutrality will profoundly affect the development and utilization of China's existing energy resources.we also need to explore and discover more ways to absorb carbon dioxide.

Guizhou is located in the core area of karst area, and its natural environment and social development are typical. Karst distribution area accounts for 73.79% of the total land area of the province. Guizhou is also an important coal and phosphate industry base in China, which is a key area of carbon emissions.Driven by the ambitious goal of carbon peak and carbon neutrality, based on the characteristics of karst areas, this study summarizes the domestic and foreign ecological restoration and negative carbon, water-energy-carbon coupling and energy saving and emission reduction, industrial energy saving and emission reduction technologies in karst areas, and carbon dioxide capture, utilization and storage (CCUS) technologies.Starting from the green development environment, on the basis of systematic analysis of the advantages, disadvantages, opportunities and challenges of Guizhou's medium and long-term development, this paper studies and judges the trend and orientation of Guizhou's green development, and combines with the goal of carbon neutrality. This paper puts forward some countermeasures, such as insisting on clean and efficient low-carbon utilization, strengthening the research and development of carbon emission reduction technology and the construction of carbon sink capacity, and puts forward some specific suggestions, such as strengthening top-level design, defining the path of emission reduction, guiding the transformation of enterprises, and ensuring science and technology and talents.It is hoped that these strategic proposals will help Guizhou develop with high quality and achieve the goal of double carbon as soon as possible.

4..Chapter 2 adds nothing significant to the topic discussed. This is journalistic content that does not indicate the scientific value of this work. I propose to delete it or radically shorten it, pointing only to the significant development trends that the Author intends to discuss in the review of the thematic literature.

5. Chapter 2 should be drafted as comments on the thematic publications cited. There are no publications relevant to the topic in this chapter.

The second chapter is rewritten, which investigates the current situation of energy and carbon neutrality in Guizhou and analyzes the data model, and also predicts the economy and future development, providing support for the subsequent selection of corresponding methods and technologies.

6. Does Figure 4 show the authors' research or is it based on data from other publications?

Deleted. The map is a comprehensive reference of multiple data. There may be a problem

  1. "Case Analysis of Green and Carbon Neutral Coordinated Development In GuiZhou" - the title of chapter 4 is misunderstood - what the Authors understand by the term natural development.

  1. Chapters 3 and 4 too extensive descriptively of a journalistic nature. It should be specified and made scientific in nature.

  1. In Chapter 5, conclusions important for the subject should be formulated.

According to the suggestions of several reviewers, the structure of the article has been greatly adjusted:

1. Introduce the basic situation of the study

2. The development and economy of energy and carbon neutrality in Guizhou were investigated and analyzed.

3. Summarize and analyze the existing advanced innovative technologies

4. Summarize the existing technology/case

5. Discuss and put forward shortcomings and problems

6. Suggest, give strategies and solutions

Reviewer 2 Report

The green industrilization and carbon neutralization are daily topics. However, the study has critical issues and needs major amendments as explained below:

1. The aim of the study is not clear, so it should be revized by the what the authors do, why, how. 

2. The abstract mainly summurization of background but study related findings is missing, so please re-write the abract

3. Is this review article or about the discussion avaliable strateties and suggest new strategies? This idea has been made unclear the manuscript. Please be conspicuous.

4. The methodology part should be added in the MS, what kind of literature can be inclueded and exclueded in the study.

5. In the part3: subsections are the literature review or authours findings after literature search...these kind of confusions are through the MS. please identify these kind of confusions

6. A chart about dilemmas and problems, and their solution suggestions can be helfpull.

7. There is a many case studies in the MS as part 4. however, they seem like a promotion sheet, what kind of problem solved by these case practise? what they aimed, what qualitative and quantitative profits in the green industrials and carbon neutrulization strategies.

8. A diagram can be generate for the suggestion and their end points in the topic.

Author Response

Dear reviewer:

Thank you very much for your work.We have learned much from your and other two reviewers’ comments,which are fair,encouraging and constructive.After carefully studying the comments and your advice, we have made corresponding changes.

1.According to the suggestions of the reviewers, the title is optimized.

The new title is:Innovation Strategy of Green Development and Carbon Neutralization in Guizhou — An Overview

2.The aim of the study is not clear, so it should be revized by the what the authors do, why, how.

This problem does exist. The structure of the article has been greatly adjusted, and the abstract and introduction have been rewritten.

3.Is this review article or about the discussion avaliable strateties and suggest new strategies? This idea has been made unclear the manuscript. Please be conspicuous.

The article is through the existing policies and technical methods of the summary and combing, on the basis of the review, combined with the current situation in Guizhou, put forward some suggestions and thinking, the original text of the article to make structural adjustment, clear this point.

4. The methodology part should be added in the MS, what kind of literature can be inclueded and exclueded in the study.

5. In the part3: subsections are the literature review or authours findings after literature search...these kind of confusions are through the MS. please identify these kind of confusions

The second and third chapter is rewritten, which investigates the current situation of energy and carbon neutrality in Guizhou and analyzes the data model, and also predicts the economy and future development, providing support for the subsequent selection of corresponding methods and technologies.

6.There is a many case studies in the MS as part 4. however, they seem like a promotion sheet, what kind of problem solved by these case practise? what they aimed, what qualitative and quantitative profits in the green industrials and carbon neutrulization strategies.

According to the suggestions of several reviewers, the structure of the article has been greatly adjusted:

1. Introduce the basic situation of the study

2. The development and economy of energy and carbon neutrality in Guizhou were investigated and analyzed.

3. Summarize and analyze the existing advanced innovative technologies

4. Summarize the existing technology/case

5. Discuss and put forward shortcomings and problems

6. Suggest, give strategies and solutions

7.A diagram can be generate for the suggestion and their end points in the topic.

Added fig 8

See the revised draft for more revisions.

Reviewer 3 Report

This paper summarizes and analyses green technologies such as "ecological restoration and negative carbon in karst areas, water-energy-carbon coupling and energy saving and emission reduction, industrial energy saving and emission reduction technologies in karst areas, and carbon dioxide capture, utilization and storage (CCUS) in karst areas".

In this paper, there is no quantitative reference by which the recommendations of the authors can be examined. There is no economic analysis indicating the cost of the proposals. I would expect to see a statistical and economic analysis and even make predictions that would confirm the authors' proposals.

The paper lacks a summary, conclusions, and recommendations.  The conclusions will be based on what was done by the authors.

Here are my detailed comments:

1)      Fig. 2 (should be changed to Figure 2.) is not understood how it relates to the statement that - "The limitation and scarcity of energy resources, as well as their strategy and security, determine that the comprehensive utilization of energy and energy conservation and emission reduction must be supported by scientific and technological progress. From where Fig. 2 It is taken why there is no reference to this in the bibliography?

2)      The paper requires rewriting and editing.

3)       Add to the abstract adequate detail and quantitative detail of the incremental improvement your proposal offers against the state of the art.

4)       Make the exposition tight and clear (right now it is meandering and redundant, particularly in the first part of the paper).

5)       Make clear what is new, what are gaps in the literature being addressed, and why doing so is important.

6)       In my opinion, the background is not well organized. It would be better if the authors first explain the motivation for their study, then discuss the problem statement, and finally review the relevant studies.

7)       The research objectives and methodology should be better explained and motivated.

8)     The authors must indicate what does it paper add to the subject area compared with other published material.

9)   The objectives must be clearly indicated in the abstract.

10)    In the conclusions section, the authors should provide a general interpretation of the rustles, the unique contributions of the paper, and the limitations of the research's managerial implications.

11)    The authors must add updated articles 3 to 5 references from the journal "sustainability ".

I do hope you find these comments and questions helpful in improving the manuscript.

Author Response

Dear reviewer two:

Thank you very much for your work.We have learned much from your and other two reviewers’ comments,which are fair,encouraging and constructive.After carefully studying the comments and your advice, we have made corresponding changes.

1.According to the suggestions of the reviewers, the title is optimized.

The new title is:Innovation Strategy of Green Development and Carbon Neutralization in Guizhou — An Overview

  1. The paper requires rewriting and editing.

According to the suggestions of several reviewers, the structure of the article has been greatly adjusted:

1. Introduce the basic situation of the study

2. The development and economy of energy and carbon neutrality in Guizhou were investigated and analyzed.

3. Summarize and analyze the existing advanced innovative technologies

4. Summarize the existing technology/case

5. Discuss and put forward shortcomings and problems

6. Suggest, give strategies and solutions

3.Add to the abstract adequate detail and quantitative detail of the incremental improvement your proposal offers against the state of the art.

We rewrite this part:Abstract:"Carbon peak in 2030 and carbon neutrality in 2060" is China's major strategic decision and development needs. Driven by the ambitious goal of carbon peak and carbon neutralization, the development of green innovation technology is an important way to achieve the goal. How to speed up the energy transformation and how to tap the carbon sink capacity of the natural ecosystem is the key. The strategic path of green development deserves further discussion. This study takes Guizhou Province as an example.Based on the actual situation of Guizhou Province and the characteristics of karst areas, through the collection and collation of existing literature, policies and technologies, and the analysis of typical cases, this paper summarizes and analyzes green technologies such as ecological restoration and negative carbon in karst areas, water-energy-carbon coupling and energy saving and emission reduction, industrial energy saving and emission reduction technologies in karst areas, and CCUS technology for carbon dioxide capture, utilization and storage.On this basis, the trend and orientation of green development in Guizhou are studied and judged, and the countermeasures such as adhering to clean and efficient low-carbon utilization, strengthening the research and development of carbon emission reduction technology and carbon sink capacity building are put forward, and the key core technology research and development innovation is put forward to establish a low-carbon science and technology innovation system. Promote efficient use of energy, recycling of resources, negative emissions and other fields. Specific suggestions such as accelerating the transformation and application of green and low-carbon scientific and technological achievements.

4.Make the exposition tight and clear (right now it is meandering and redundant, particularly in the first part of the paper).

  1. Make clear what is new, what are gaps in the literature being addressed, and why doing so is important.

This problem does exist. The structure of the article has been greatly adjusted, and the abstract and introduction have been rewritten.

The article is through the existing policies and technical methods of the summary and combing, on the basis of the review, combined with the current situation in Guizhou, put forward some suggestions and thinking, the original text of the article to make structural adjustment, clear this point.

The second and third chapter is rewritten, which investigates the current situation of energy and carbon neutrality in Guizhou and analyzes the data model, and also predicts the economy and future development, providing support for the subsequent selection of corresponding methods and technologies.

6.Figure 2 refers to the IEA 2021 report, 34 in Ref.

See the revised draft for more revisions.

Round 2

Reviewer 1 Report

Dear Author

The corrections made improved the manuscript. I accept for publication

Greetings

Reviewer

Author Response

Thank you again for your work. I wish you success in your work and a happy life

Reviewer 2 Report

The authors have been revised the MS. However, the some English errors and some connection errors between the statements detected . Therefore, careful reading is needed. 

Author Response

We have polished the English expression of the article and edited it in English before the official mdpi.Thank you again for your work. I wish you success in your work and a happy life.

Reviewer 3 Report

Accept in present form

Author Response

We have polished the English expression of the article and edited it in English before the official mdpi.Thank you for your work. I wish you success in your work and a happy life.